# Genes Involved in DNA Repair and Mitophagy Protect Embryoid Bodies from the Toxic Effect of Methylmercury Chloride under Physioxia Conditions

**DOI:** 10.3390/cells12030390

**Published:** 2023-01-21

**Authors:** Justyna Augustyniak, Hanna Kozlowska, Leonora Buzanska

**Affiliations:** 1Department of Neurochemistry, Mossakowski Medical Research Institute, Polish Academy of Sciences, 02-106 Warsaw, Poland; 2Laboratory of Advanced Microscopy Technique, Mossakowski Medical Research Institute, Polish Academy of Sciences, 02-106 Warsaw, Poland; 3Department of Stem Cell Bioengineering, Mossakowski Medical Research Institute, Polish Academy of Sciences, 02-106 Warsaw, Poland

**Keywords:** human induced pluripotent stem cells, embryoid bodies, methylmercury chloride, mitochondria, oxygen concentration, DNA damage and repair, developmental toxicity, neurodevelopmental toxicity

## Abstract

The formation of embryoid bodies (EBs) from human pluripotent stem cells resembles the early stages of human embryo development, mimicking the organization of three germ layers. In our study, EBs were tested for their vulnerability to chronic exposure to low doses of MeHgCl (1 nM) under atmospheric (21%O_2_) and physioxia (5%O_2_) conditions. Significant differences were observed in the relative expression of genes associated with DNA repair and mitophagy between the tested oxygen conditions in nontreated EBs. When compared to physioxia conditions, the significant differences recorded in EBs cultured at 21% O_2_ included: (1) lower expression of genes associated with DNA repair (*ATM*, *OGG1*, *PARP1*, *POLG1*) and mitophagy (*PARK2*); (2) higher level of mtDNA copy number; and (3) higher expression of the neuroectodermal gene (*NES*). Chronic exposure to a low dose of MeHgCl (1 nM) disrupted the development of EBs under both oxygen conditions. However, only EBs exposed to MeHgCl at 21% O_2_ revealed downregulation of mtDNA copy number, increased oxidative DNA damage and DNA fragmentation, as well as disturbances in *SOX17* (endoderm) and *TBXT* (mesoderm) genes expression. Our data revealed that physioxia conditions protected EBs genome integrity and their further differentiation.

## 1. Introduction

Methylmercury (MeHg) is an environmental toxin that accumulates primarily in fish and seafood [1,2,3]. In humans, methylmercury can be transferred directly to the fetus through the placenta. After crossing the placenta, it disturbs the normal development of the central nervous system [4]. The mechanism of methylmercury toxicity involves four processes leading to genotoxicity: the generation of free radicals and oxidative stress, the action on microtubules, the influence on DNA repair mechanisms, and direct interaction with DNA molecules [5]. The consequences of exposure of human embryos to methylmercury compounds can be associated with teratogenesis that manifests in congenital malformations [4].

Human pluripotent stem cells (hPSCs), which include human embryonic stem cells (hESCs) and human induced pluripotent stem cells (hiPSCs), are characterized by the ability to self-renew and become almost any cell type in the human body [6,7]. Hence, hPSCs provide an opportunity to gain insights into human embryonic development [8,9,10]. To maintain the ability to differentiate into many cell types without propagating DNA errors, pluripotent stem cells (PSCs) must protect their genome integrity. Compared to somatic cells, PSCs are characterized by increased expression of DNA repair genes [11,12]. A large group of DNA repair genes expressed in PSCs is involved in homological recombination (HR) [13].

When human pluripotent stem cells (hPSCs, hiPSCs, hESCs) grow in suspension without feeder layers, they spontaneously form aggregates known as embryoid bodies (EBs) [14,15]. EBs resemble self-organizing aggregates that reflect some aspects of early embryogenesis [16,17]. During EBs formation and development, PSCs transition from homogeneous epithelial-like cells into a three-dimensional organization of various cell types with multifaceted cell–cell interactions and lumen formation [17,18]. The process resembles gastrulation due to the polarization and regionalization of tissues. Gastrulation leads to the formation of the three germ layers of the embryos, the ectoderm, the mesoderm, and the endoderm [9,15,19,20,21]. EBs formation and development processes are characterized by changes in the gene expression profile corresponding to sequential stages of embryonic development [22]. EBs development reflects embryo development processes such as symmetry breaking, asymmetric gene expression, and approximate axis formation and elongation [17,23]. On days 13–14 of human embryo development, all markers of the three germ layers are expressed [24,25,26]. Therefore, human PSCs and EBs provide valuable models for studying the embryotoxicity and teratogenicity of drugs and environmental toxins in vitro [27,28,29,30,31,32,33,34].

One of the critical factors influencing cell characteristics and function is the oxygen concentration in the microenvironment [35,36]. In human embryos, embryonic stem cells (ESCs) reside in the inner cell mass (ICM) of a blastocyst under physiologically low oxygen conditions (physioxia), defined by the partial oxygen pressure (3–5% oxygen, 23–38 mm Hg PO_2_) [37]. Atmospheric oxygen culture conditions, but not 5% O_2_, were shown to negatively influence human embryo development rates [38]. This effect depends on the developmental stage and is crucial in the early stages of embryo development [39]. Furthermore, physioxia (5% O_2_) was shown to positively influence the maintenance and self-renewal capacity of hiPSCs pluripotency and somatic cell reprogramming [40,41,42]. The atmospheric oxygen level can increase mitochondrial OxPhos activity with higher reactive oxygen species (ROS) production, which can induce spontaneous differentiation in hiPSCs and hESCs [43]. OxPhos activation is implemented in the initial stages of PSC differentiation [44,45]. Many studies have shown that low oxygen (hypoxia) levels reduce the expression of DNA repair genes in cancer cells [46,47,48,49]; however, the effect of physioxia (5%) on the expression of EBs DNA repair genes has not yet been evaluated.

Teratogenesis and neurodevelopmental deficits caused by developmental toxins are related to DNA oxidative damage and damage repair disturbance [50]. Correct repair of DNA damage during the early stages of embryonic development is essential for proper development, control of germ layer commitment, and further embryonic fate [51,52,53]. In ESCs, comprehensive DNA damage repair mechanisms are responsible for preserving and maintaining their genome integrity [11]. However, another important mechanism that protects the integrity of the mitochondrial genome is the selective degradation of mitochondria by autophagy (mitophagy), which prevents the accumulation of dysfunctional mitochondria [54,55]. In the case of excessive DNA damage and repair dysfunction, the proapoptotic pathway is activated, leading to cell death and embryo mortality [56].

Our previous study [29] showed that oxygen was an essential modulator in the MeHgCl mechanism of toxicity in undifferentiated hiPSCs. In this report, we concentrated on the toxic effect of MeHgCl on EBs formation and their neuroectodermal fate, depending on different high (21%) or low (5%) oxygen concentrations. First, our experimental design was to establish the relative expression of genes involved in DNA damage repair and mitophagy in EBs cultured under different oxygen conditions. Second, we established the sensitivity of EBs to MeHgCl under standard conditions for cell culture, e.g., at a 21% O_2_ concentration, to determine the highest dose capable of maintaining EB integrity and proper morphology. Finally, we compared the vulnerability of EBs (during differentiation into three germ layers) to a low dose of MeHgCl (1 nM) exposure under two tested oxygen conditions. **By comparing how exposure to MeHgCl affected DNA damage, mtDNA content, expression of DNA repair genes, and selected developmental genes, physioxia (5%) was shown to be a protective condition for EB neurodevelopmental capacity and cell genome integrity.**

## 2. Materials and Methods

The first part of the experiments was carried out on undifferentiated hiPSCs cultured under atmospheric oxygen (21%) and EBs derived from hiPSCs. EBs were formed before MeHgCl (Sigma-Aldrich, Merck KGaA, Darmstadt, Germany) treatment. After adding MeHgCl, the EBs were cultured at atmospheric (21%) and/or reduced oxygen concentrations (5%). The in vitro models (hiPSCs in monolayers and EBs) and tests used in this study are presented below.

### 2.1. MeHgCl Toxicity for hiPSCs Cultured in Atmospheric Oxygen Conditions

#### 2.1.1. hiPSCs Culture and hiPSCs MeHgCl Treatment in Monolayer

The hiPSCs (Gibco^®^ Human Episomal iPSC Line, Life Technologies, (Thermo Fisher Scientific, Waltham, MA, USA) in the undifferentiated stage were cultured at 21% O_2_ in Essential 8 medium (Thermo Fisher Scientific, Waltham, MA, USA) in a 6-well plate covered with rh-vitronectin (Thermo Fisher Scientific, Waltham, MA, USA) [57]. For experiments, hiPSCs cultured at 21% O_2_ were seeded in 6- or 96-well plates (Nunc, Thermo Fisher Scientific, Waltham, MA, USA) covered with rh-vitronectin (Thermo Fisher Scientific, Waltham, MA, USA) in Essential 8 medium. The next day, MeHgCl (Sigma-Aldrich, Merck KGaA, Darmstadt, Germany) was added to hiPSCs at different concentrations (0–1 µM). hiPSCs exposure to MeHgCl was continued for the next 24 h. At this time, apoptosis, ROS accumulation, mitochondrial membrane potential, and the ability to form EBs from MeHgCl-treated hiPSCs were evaluated.

#### 2.1.2. Apoptosis Level Assay in hiPSC Monolayer Culture

The hiPSCs were cultured in Essential 8 Medium without or with MeHgCl at a dose of 0–1 µM for 24 h. The next day, 50 µg/mL propidium iodide (PI) (Sigma-Aldrich, Merck KGaA, Darmstadt, Germany) was added to the medium where the hiPSCs were placed. After 20 min of incubation with PI, the hiPSCs were analyzed. Fluorescence was measured at 485–538 nm wavelength with a Fluoroscan Ascent FL plate reader (Labsystems, Tiilitie, Vantaa, Finland)). The results are presented as a percentage (%) of the control.

#### 2.1.3. Alamar Blue Cell Viability Assay in hiPSC Monolayer Culture

The hiPSCs were cultured in Essential 8 Medium without or with MeHgCl (Sigma-Aldrich, Merck KGaA, Darmstadt, GermanyMerck) at a dose of 0–1 µM for 24 h. After 24 h of exposure, hiPSCs were incubated at 37 °C with Alamar Blue (0.1 mg/mL) (Sigma-Aldrich, Merck KGaA, Darmstadt, Germany) diluted in a culture medium (1:10) for 3 h. The fluorescence intensity of the REDOX indicator was measured at 544 nm and 590 nm for excitation and emission, respectively, using a Fluoroscan Ascent FL plate reader (Labsystems, Tiilitie, Vantaa, Finland) [29]. The number of living cells was calculated on the calibration curve. The results are presented as a percentage (%) of the control.

#### 2.1.4. ROS Accumulation Assay in hiPSC Monolayer Culture

The hiPSCs were cultured in Essential 8 Medium without or with MeHgCl at a dose of 0–1 µM for 24 h. After 24 h of hiPSC exposure to MeHgCl, DCFH-DA (Sigma-Aldrich, Merck KGaA, Darmstadt, Germany) was added to the cells at a concentration of 1 µM. The hiPSCs were incubated with DCFH-DA for 1 h and washed three times with PBS. Fluorescence was measured in OptiMem Medium (Life Technologies, Thermo Fisher Scientific, Waltham, MA, USA) at 485–538 nm wavelength with a Fluoroscan Ascent FL plate reader (Labsystems, Tiilitie, Vantaa, Finland) [29]. The results are presented as a percentage (%) of the control.

#### 2.1.5. Mitochondrial Membrane Potential Assay in hiPSC Monolayer Culture

The hiPSCs were cultured in Essential 8 Medium without or with MeHgCl (Sigma-Aldrich, Merck KGaA, Darmstadt, GermanyMerck)at a dose of 0–1 µM for 24 h. After 24 h of hiPSC incubation with MeHgCl, MitoTracker^®^ Red CMXRos (Thermo Fisher Scientific, Waltham, MA, USA) at a concentration of 50 nM was added to the hiPSCs. After 1 h, the cells were washed three times with PBS (Sigma-Aldrich, Merck KGaA, Darmstadt, Germany), and fluorescence was measured at 544–590 nm. The level of fluorescence intensity in the samples was tested relative to controls using a Fluoroscan Ascent FL plate reader (Labsystems, Tiilitie, Vantaa, Finland) [29]. The results are presented as a percentage (%) of the control.

#### 2.1.6. hiPSCs Ability to Form EBs after MeHgCl Treatment

hiPSCs (Gibco^®^ Human Episomal iPSC Line, Life Technologies, Thermo Fisher Scientific, Waltham, MA, USA) were seeded on a 6-well plate covered with rh-vitronectin (Thermo Fisher Scientific, Waltham, MA, USA) in Essential 8 Medium (Thermo Fisher Scientific, Waltham, MA, USA) [17]. On the next day, MeHgCl at different concentrations (0–1 µM) was added. After 24 h of MeHgCl (Sigma-Aldrich, Merck KGaA, Darmstadt, Germany) exposure, hiPSCs were detached with 0.5 mM EDTA (Sigma-Aldrich, Merck KGaA, Darmstadt, Germany) and transferred to 6-well anti-adhesive plates (Nunc, Thermo Fisher Scientific, Waltham, MA, USA). EBs growth was monitored for the next 7 days. EBs were formed in Essential 8 Medium without MeHgCl. The images were obtained with an AxioCam MRC5 camera and an Axiovert 200 fluorescence microscope (Carl Zeiss, Gottingen, Germany).

### 2.2. MeHgCl Toxicity in hiPSC-Derived EBs Cultured under Atmospheric Conditions

#### 2.2.1. EBs Generation and MeHgCl Treatment under 21% Oxygen Conditions

Before EBs were formed from hiPSCs, hiPSCs were grown in Essential 8 Medium (Thermo Fisher Scientific, Waltham, MA, USA) for 2 weeks on a 6-well plate covered with rh-vitronectin (Thermo Fisher Scientific, Waltham, MA, USA) in 21% O_2_. After that period, the hiPSCs were detached from the plate with 0.5 mM EDTA (Sigma-Aldrich, Merck KGaA, Darmstadt, Germany). The detached hiPSCs colonies were transferred to 6 wells of antiadhesive plates (Nunc, Thermo Fisher Scientific, Waltham, MA, USA) to form EBs [17]. For the next 3 days, the EBs were cultured in suspension in Essential 8 Medium. Afterward, the Essential 8 medium was changed to Essential 6 Medium (Thermo Fisher Scientific, Waltham, MA, USA) to induce spontaneous differentiation of EBs into the three germ layers. On the day of switching from Essential E8 medium to Essential E6 medium, MeHgCl (Sigma-Aldrich, Merck KGaA, Darmstadt, Germany) was added to the EBs. EBs were cultured in Essential 6 Medium with or without MeHgCl for the next 14 days. MeHgCl at different concentrations (0 µM to 0.5 µM) was added to EBs growth in 21% O_2_ in Essential E6 Medium for 14 days. Finally, the changes in EBs morphology and cell apoptosis were analyzed.

#### 2.2.2. Analysis of EB Morphology Changes under MeHgCl Treatment

Newly formed EBs were induced for spontaneous differentiation by Essential E6 Medium (Thermo Fisher Scientific, Waltham, MA, USA) for 14 days in 21% O_2_ [17] and simultaneously treated with MeHgCl (Sigma-Aldrich, Merck KGaA, Darmstadt, Germany) at different concentrations (0 µM to 0.5 µM). EBs morphology after MeHgCl treatment was analyzed after different periods of exposure at 4, 7, and 14 days with light microscopy (Axiovert 40C, Carl Zeiss, Gottingen, Germany).

#### 2.2.3. Visualization of Apoptosis in Suspension-Cultured EBs

EBs were cultured in Essential 6 Medium (Thermo Fisher Scientific, Waltham, MA, USA) without or with MeHgCl (Sigma-Aldrich, Merck KGaA, Darmstadt, Germany) at doses of 0–0.5 µM for 24 h. The next day, propidium iodide (PI) (Sigma-Aldrich, Merck KGaA, Darmstadt, Germany) was added to the medium at a 50 µg/mL concentration. After 20 min incubation with PI, apoptosis was analyzed with a fluorescence microscope. Images were obtained with an Axiovert 200 fluorescence microscope (Carl Zeiss, Gottingen, Germany).

### 2.3. Toxicity of a Chronic Low Dose of MeHgCl in hiPSC-Derived EBs Cultured under Atmospheric and Low Oxygen Conditions

#### 2.3.1. EBs Generation and MeHgCl Treatment under 21% and 5% Oxygen Conditions

Before EBs were formed, hiPSCs were grown in Essential 8 Medium (Thermo Fisher Scientific, Waltham, MA, USA) for 2 weeks on a 6-well plate covered with rh-vitronectin (Thermo Fisher Scientific, Waltham, MA, USA) in 21% or 5% oxygen. Next, the hiPSCs were detached from the plate with 0.5 mM EDTA. The detached hiPSCs colonies were transferred to 6 wells of antiadhesive plates (Nunc) to form EBs [17]. For the next 3 days, the EBs were cultured in suspension in Essential 8 Medium. Afterward, the Essential 8 medium was replaced with Essential 6 Medium (Thermo Fisher Scientific, Waltham, MA, USA) to induce spontaneous differentiation of EBs into the three germ layers. On switching from Essential E8 medium to Essential E6 medium, MeHgCl (Sigma-Aldrich, Merck KGaA, Darmstadt, Germany) was added to EBs at a concentration of 1 nM. EBs were cultured in Essential 6 Medium with or without MeHgCl for the next 14 days. After that period, the EBs samples were collected and frozen at −80 °C. Frozen EBs samples were used for future analyses, such as (1) DNA damage (AP sites), (2) mtDNA copy number, (3) gene expression, and (4) immunocytochemical analysis. The methods used for this analysis are presented below.

#### 2.3.2. Total Genomic DNA Isolation

Genomic DNA was isolated from EBs grown in 21% and 5% oxygen concentrations from control and experimental cultures. EBs were treated with MeHgCl (Sigma-Aldrich, Merck KGaA, Darmstadt, Germany) for 14 days before DNA isolation. Genomic mini kit (A & A Biotechnology, Gdynia, Poland) was used to isolate DNA. The quantity and purity of total genomic DNA were determined by spectrometry using a DS-11 FX spectrophotometer (Denovix, Wilmington, Denmark). The purified genomic DNA was stored at −20 °C until further analysis.

#### 2.3.3. DNA Damage Assay

Genomic DNA damage was quantified with an apurinic/apyrimidinic (AP) site colorimetric assay (Abcam, Cambridge, UK) according to the manufacturer’s protocol [58]. Optical density at a wavelength of 650 nm was measured with an OMEGA (BMC Labtech, Ortenberg, Germany) plate reader.

#### 2.3.4. Total RNA Isolation

RNA was isolated from EBs grown in 21% or 5% oxygen after 14 days of exposure to 1 nM MeHgCl with a Total RNA mini kit (A & A Biotechnology, Gdynia, Poland). A Clean-Up RNA Concentrator Kit (A & A Biotechnology, Gdynia, Poland) was used to remove genomic DNA from RNA samples. The RNA concentration was determined using a DS-11 FX spectrophotometer (Denovix, Wilmington, Denmark).

#### 2.3.5. Reverse Transcription (RT)

cDNA was obtained from total RNA in a reverse transcription-polymerase chain reaction (RT) with a High-Capacity RNA-to-cDNA™ Kit (Applied Biosystems, Thermo Fisher Scientific, Waltham, MA, USA).

#### 2.3.6. Quantitative Polymerase Chain Reaction (qPCR)

The mtDNA copy numbers were quantified using the qPCR method [29]. qPCR was performed with 10 ng of DNA template in 15 μL of reaction mixture containing 12.5 μL of iTaq™ Universal SYBR^®^ Green Supermix (Bio-Rad, Hercules, CA, USA) and 0.25 μM forward and reverse primers on a LightCycler 96 (Roche, Basel, Switzerland). qPCR was subjected to a hot start at 95 °C for 3 min followed by 45 cycles of denaturation at 95 °C for 10 s, annealing at 60 °C for 30 s, and extension at 72 °C for 30 s in Roche 96-well qPCR plates (Roche, Basel, Switzerland) with foils (BLIRT S.A., Gdansk, Poland). To determine the relative mitochondrial DNA content, the following equation was used: relative mitochondrial DNA content = 2 × 2^ΔCT^, where ΔCT was calculated from the formula (ΔCT = nDNA_CT_ − mtDNA_CT_) [59]. The primer sequences used for qPCR in these studies are presented in Table 1.

#### 2.3.7. qRT-PCR

For qRT-PCR, 10 ng of cDNA was loaded with 0.25 µM forward and reverse primers and 12.5 μL of iTaq™ Universal SYBR^®^ Green Supermix (Bio-Rad, Hercules, CA, USA) in Roche 96-well qPCR plates with foils (BLIRT S.A., Gdansk, Poland) in the following steps: initial denaturation step at 95 °C for 3 min, 45 cycles of denaturation at 95 °C for 10 sec, and annealing/extension at 58 °C for 1 min on a LightCycler 96 (Roche, Basel, Switzerland) [29]. *EID2* was used as a housekeeping gene to calculate relative gene expression. Relative gene expression was measured in samples isolated from EBs cultured without MeHgCl and EBs exposed to 1 nM MeHgCl (Sigma-Aldrich, Merck KGaA, Darmstadt, Germany) under different oxygen conditions (21% and 5%) for 14 days. The housekeeping gene was predicted in NormFinder software. The primer sequences used to validate reference genes were previously described by Augustyniak et al. [60]. The primer sequences used for RT-qPCR in these studies are presented in Table 2 and Table 3.

#### 2.3.8. Immunofluorescence Staining

EBs were collected and transferred to a 15 mL conical tube on day 14 of spontaneous differentiation. When the aggregates of EBs sank to the bottom of the tube, the medium was removed, and the EBs were fixed with 4% paraformaldehyde (PFA) (Sigma-Aldrich, Merck KGaA, Darmstadt, Germany) solution in PBS (Sigma-Aldrich, Merck KGaA, Darmstadt, Germany) for 30 min at room temperature, washed with PBS, and placed in serial dilutions of PBS buffered sucrose solutions (10, 20, and 30%, in sequence) (Sigma-Aldrich, Merck KGaA, Darmstadt, Germany) at 25–28 °C. Each solution was replaced every 30 min. EBs were embedded in cryo-embedding media (Tissue-Tek^®^ OCT compound) (Sakura, Torrance, CA, USA) and frozen at −80 °C. Materials were sectioned on a cryostat, and EBs slides were washed with PBS. Next, the slides were permeabilized with 0.1% Triton X-100 Sigma-Aldrich, Merck KGaA, Darmstadt, Germany) and blocked with 10% goat serum (Thermo Fisher Scientific, Waltham, MA, USA). The following primary antibodies were applied overnight: early neuroectoderm marker: PAX6, neural marker: NESTIN; endoderm marker: FOXA2, mesoderm marker: αSMA; neuronal marker NF200; pluripotency markers: OCT4, NANOG; proliferation marker: Ki67; DNA damage marker: Histone H2A.X phosphorylated at Ser139 and the apoptosis marker CASP3. After washing the slides with PBS, the appropriate secondary antibodies (Alexa Fluor 488 or 546) were applied for 1 h, and cell nuclei were counterstained with Hoechst 33258 (Sigma-Aldrich, Merck KGaA, Darmstadt, Germany)). After the final wash, the slides were mounted in Fluorescent Mounting Medium (Dako, Jena, Germany) [61]. As a control, the primary antibodies were omitted during immunofluorescence staining. A confocal laser scanning microscope LSM 780/Elyra PS.1 (Carl Zeiss, Jena, Germany) was used to obtain detailed images of the cells. After the acquisition, the images were processed using ZEN 2012 SP5 software (Carl Zeiss, Munich, Germany). All ICF images were obtained from the Laboratory of Advanced Microscopy Techniques of the Mossakowski Medical Research Institute, Polish Academy of Sciences. The primary antibodies used in the study are presented in Table 4 and Table 5.

#### 2.3.9. Gene Function Prediction and Network Analysis

The GeneMANIA (https://genemania.org/ (accessed on 10 June 2022, 25 October 2022, 27 October 2022) web tool was used for gene function prediction and network analysis. Network analysis and functional enrichment analysis were performed for the proteins for which immunocytochemical staining (ICC) was performed and for the genes for which gene expression analysis (qRT-PCR) was performed in this study.

### 2.4. Statistical Analysis

The results were analyzed in GraphPad Prism 5.0 (Insight Partners, New York, NY, USA) using the following statistical tests: (1) the t-Mann–Whitney U test; (2) the Kruskal-Wallis test with Dunn’s multiple comparison test; and (3) the two-way ANOVA with Bonferroni posttests. The experiments were carried out three times with three replicates. The figures present the data as the mean ± SD. The significance of the obtained results is presented as *p* < 0.05 (*); *p* < 0.001 (**); *p* < 0.0001 (***).

## 3. Results

### 3.1. The Effect of MeHgCl on hiPSCs Grown in 21% O_2_

The schedule of the experimental design to assess the toxicity of different doses of MeHgCl for hiPSCs growing in a monolayer in 21% O_2_ (Figure 1B–F) and their ability to form EBs after detachment (Figure 1G) is shown in Figure 1A. The results are summarized in Appendix A.

#### 3.1.1. Apoptosis

Apoptosis was assessed in hiPSCs grown in a monolayer under atmospheric oxygen by PI staining. Quantitative data measured as the fluorescence intensity of PI-positive cells revealed an increase in apoptotic cells in hiPSCs treated with MeHgCl (24 h) compared to hiPSCs cultured without MeHgCl. The increase was statistically significant (*p* < 0.01) only in hiPSCs exposed to MeHgCl at a dose of 1 µM (140% ± 9.82), as shown in Figure 1C. The number of apoptotic cells increased by 15.6% ± 13.20 in hiPSCs cultured in medium with the addition of 0.5 µM MeHgCl. For comparison, the presence of 0.25 µM MeHgCl in the medium increased the number of apoptotic cells by 22.4 ± 25.82, and the differences were not statistically significant.

#### 3.1.2. Viability

The viability of hiPSCs grown under atmospheric oxygen was evaluated with Alamar blue containing resazurin, which measured colorimetric changes in response to the reduction in cellular metabolism. The most evident change in hiPSC viability was observed in the cells exposed to MeHgCl at 1 µM. The viability of hiPSCs treated with 1 µM MeHgCl decreased to 78.41% ± 5.67 compared to the hiPSCs untreated with MeHgCl (Figure 1D). Exposure to MeHgCl was carried out for 24 h. The changes in hiPSCs viability under the tested conditions were statistically insignificant.

#### 3.1.3. ROS Accumulation

In hiPSCs cultured with MeHgCl in 21% O_2_, the accumulation of ROS relative to the control (%) was analyzed with DCFH-DA (Figure 1B,E). MeHgCl at different doses ranging from 0 µM to 1 µM was added to the cells for 24 h. In hiPSCs, MeHgCl significantly (*p* < 0.05) increased ROS accumulation only at a dose of 1 µM (145.6% ± 5.07). For other MeHgCl doses, the changes in ROS accumulation detected in the hiPSCs culture were insignificant.

#### 3.1.4. Mitochondrial Membrane Potential

After 24 h of hiPSCs exposure to MeHgCl (0–1 μM), the mitochondrial membrane potential (Δψ m) of hiPSCs was tested with MitoTracker red (Figure 1B,F). An insignificant effect of MeHgCl at doses of 0–0.5 µM on the Δψ m was observed, and MeHgCl decreased the Δψ m when applied at all the analyzed doses. However, significant changes (*p* < 0.05) in this parameter were observed only in the hiPSCs grown in the medium with the addition of MeHgCl at a concentration of 1 µM (45.39% ± 0.55).

#### 3.1.5. The Effect of MeHgCl on hiPSC-Derived EB Formation under 21% O_2_ Conditions

The hiPSCs were cultured for 24 h with MeHgCl applied at a dose of 0–1 µM in 21% O_2_. After this timespan, the potential of hiPSCs for EB formation was analyzed (Figure 1G). The hiPSCs treated with MeHgCl showed a reduced ability to form EBs and their further development. After 24 h of incubation with 1 µM MeHgCl, the hiPSCs were observed to lose their ability to form EBs. After 24 h of incubation with MeHgCl at doses of 0.125–0.5 µM, the ability of hiPSCs to form EBs decreased. However, the efficiency of EBs generation from hiPSCs was very low. Although EBs from MeHgCl-treated hiPSCs (0.125–0.5 µM) were created, the hiPSCs potential for EBs formation was reduced at all tested MeHgCl doses. The consequence of the loss of differentiation potential by hiPSCs was the inhibition of EB formation and their degradation during a more extended culture period (Figure 1G).

### 3.2. The Effect of MeHgCl on EB Development under 21% O_2_ Conditions

The schedule of the experimental design to assess the dose- and time-dependent sensitivity of EBs to MeHgCl at 21% O_2_ is shown in Figure 2A. In the following steps, we analyzed the developmental potential of EBs cultured at different concentrations of MeHgCl (0–0.5 µM) in 21% O_2_ (Figure 2B). After detachment of hiPSCs, EBs were spontaneously formed in Essential 8 Medium. The EBs were cultured in Essential 8 medium for 72 h before transfer to Essential 6 Medium, which induced their differentiation into the three germ layers. EBs grew in Essential 6 Medium for 14 days with or without MeHgCl addition (Figure 2A). The results are summarized in Appendix A.

Staining with propidium iodide (red) in EBs cultures incubated for 24 h with different doses of MeHgCl showed MeHgCl dose-dependent cell apoptosis. After 4 days of EBs exposure to MeHgCl, a strong cytotoxic effect (100%) was detected in the EBs cultured with the addition of 0.5 µM MeHgfCl. The 7-day incubation period with MeHgCl also produced 100% cytotoxic effects detected in EBs cultured in a medium supplemented with 0.25 µM MeHgCl. EBs exposure to MeHgCl for 14 days led to 100% EBs death at all tested doses of MeHgCl. The EBs sensitivity to MeHgCl, depending on the dose and time of exposure, is shown in Figure 2B.

### 3.3. The Effect of Different Oxygen Concentrations (21%, 5%) on the Expression of Genes Involved in DNA Damage Repair, Mitochondrial Biogenesis, and Formation of the Three Germ Layers in EBs

The schedule of the experimental design to assess relative gene expression in EBs cultured at different oxygen concentrations (21%O_2_, 5% O_2_) is shown in Figure 3A. The results are summarized in Appendix A.

The relative expression of key genes controlling the formation of three germ layers (*NES, SOX17, TBXT*), microtubules (*TUBB3*), mitochondrial biogenesis (*TFAM, POLG1*), mitophagy (*PARK2*), and DNA repair (*ATM, PAPR1, OGG1*) was tested using qRT-PCR in EBs. EBs were cultured in 21% and 5% oxygen without MeHgCl in medium-induced three-germ layer formation for 14 days (Figure 3A).

In EBs developed in 21% O_2_, compared to EBs grown in 5% O_2_, the neuroectoderm (*NES*) and endoderm (*SOX17*) formation-related genes were expressed at significantly lower levels (0.12 ± 0.08- and 0.54 ± 0.04-fold changes, respectively, *p* < 0.00), while the mesoderm (*TBXT*) did not differ between EBs cultured in either oxygen condition. Furthermore, in EBs grown at physioxia, upregulation of *TUBB3* expression (2.37 ± 0.698-fold change) was observed. The expression of key genes involved in mitochondrial biogenesis (*TFAM, POLG1*) was significantly (*p* < 0.01) different in EBs cultured in 21% O_2_ and 5% O_2_. At physioxia, the EBs were characterized by a lower relative expression of *TFAM* (0.50 ± 0.06) and higher *POLG1* (1.68 ± 0.3). Significantly higher expression of the *PARK2* gene (2.82 ± 0.34) was shown in EBs cultured in 5% O_2_. Furthermore, the relative expression of genes involved in DNA repair (*ATM, PARP1, OGG1*) was significantly higher in EBs cultured in 5% O_2_ (*ATM* (4.05 ± 0.63, *p* < 0.001), *PARP1* (1.75 ± 0.19, *p* < 0.001), and *OGG1* (1.59 ± 0.16. *p* < 0.05)). The results are presented in Figure 3B–K.

### 3.4. The Effect of Different Oxygen Concentrations and a Low Dose of MeHgCl (1 nM) on the Expression of Genes Involved in DNA Damage Repair, Mitochondrial Biogenesis, and Formation of the Three Germ Layers in EBs

The schedule of the experimental design to assess the relative expression of selected genes in EBs treated with MeHgCl vs. EBs untreated with MeHgCl in 21% O_2_ and 5% O_2_ is shown in Figure 4A. The results are summarized in Appendix A.

EBs were exposed to a low dose of 1 nM MeHgCl for 14 days of development in a low (5%) or high (21%) oxygen environment. After this period, the relative expression of key genes controlling the formation of three germ layers (*NES, SOX17, TBXT*); microtubules (*TUBB3*); mitochondrial biogenesis (*TFAM, POLG1*); mitophagy (*PARK2*); and DNA repair (*ATM, PAPR1, OGG1*) was evaluated with the qRT PCR method (Figure 4). In EBs cultured in 21% O_2_, some disturbance in the expression of genes associated with the formation of three germ layers was observed, including downregulation of *SOX17* (0.77 ± 1.36-fold change, *p* < 0.05) and a significant upregulation of *TBXT* expression (1.96 ± 1.13, *p* < 0.05). In contrast, EBs cultured in a physioxia environment revealed no significant change in the expression of the analyzed three germ layer formation markers.

The expression of genes involved in mitochondrial biogenesis (*TFAM, POLG1*) was altered after MeHgCl treatment: *TFAM* (1.26 ± 0.22, *p* < 0.05) expression was significantly increased only in EBs developed in 5% O_2_, while *POLG1*, a gene involved in mtDNA replication and damage repair, was negatively regulated in EBs under both oxygen conditions. The relative expression of *POLG1* in 21% O_2_ was 0.68 ± 0.49 (*p* < 0.05), while in 5% O_2_, it amounted to 0.89 ± 0.14 (*p* < 0.05). Furthermore, the expression of *PARK2* (0.9 ± 0.74), which controls mitophagy, was significantly downregulated in EBs treated with MeHgCl at 21% O_2_ (*p* < 0.5), while in 5% O_2,_ these changes were not significant. In EBs cultures in 21% O_2_, the expression of *TUBB3* associated with microtubule organization was also altered: *TUBB3* expression increased in EBs treated with MeHgCl under atmospheric but not physioxia conditions; however, these changes were insignificant. The difference in the relative gene expression of *TUBB3* and *PARK2* in MeHgCl-treated EBs in 21% O_2_ and 5% O_2_ was significant. In EBs treated with MeHgCl, the expression of genes associated with base excision repair (BER), *PARP1* (0.84 ± 0.09) and *OGG1* (0.8 ± 0.09) were significantly (*p* < 0.01) downregulated. In EBs grown in 21% O_2_, MeHgCl treatment was not associated with any significant changes in the expression of the analyzed DNA repair genes (*ATM, PARP1, OGG1*).

Function prediction and network analysis (GeneMANIA) were performed to understand better the biological function of the analyzed gene expression changes. For this purpose, the genes were divided into two groups: genes associated with development (*NES, SOX17, TBXT, TFAM*) and those related to DNA damage repair and mitophagy (*ATM, OGG1, POLG (POLG1), PARKN (PARK2), PARP1*). The analysis of the former showed that the *SOX17* and *TBXT* genes were involved in primary germ layer formation and cell fate commitment. At the same time, *NES* and *SOX17* were associated with embryonic organ development. However, *TFAM* function was mainly associated with mitochondrial gene expression and mitochondrial RNA metabolic processes. In the predicted GeneMANIA software gene network, which included *NES, SOX17, TBXT,* and *TFAM*, the primary type of interaction was associated with physical interactions. Other interactions revealed for this gene network included co-expression, colocalization, genetic interactions, pathways, or shared protein domains (Appendix A).

The second network included the *ATM, OGG1,* and *PARP1* genes, whose main functions were to regulate the response to DNA damage stimuli and repair double-strand breaks. *POLG (POLG1)* and *OGG1* were associated with BER. Another primary function of the *POLG1* gene was related to DNA replication. The last gene analyzed was *PARKN (PARK2),* whose predicted functions were related to the regulation of reactive oxygen species, mitophagy, and mitochondrial fission. In the gene network that included *ATM, OGG1, PARP1, POLG1,* and *PARKN*, physical interaction was the primary type of interaction in the network. Other interactions between genes in the network were associated with gene co-expression, colocalization, genetic interactions, pathways, or shared protein domains (Appendix A).

### 3.5. The Effect of Different Oxygen Concentrations and a Low Dose of MeHgCl (1 nM) on mtDNA Copy Number and DNA Damage in EBs

The schedule of the experimental design to assess mtDNA copy number and DNA damage in EBs treated with MeHgCl vs. EBs untreated with MeHgCl in 21% O_2_ and 5% O_2_ is shown in Figure 5A.

#### 3.5.1. The Effect of Different Oxygen Concentrations and a Low Dose of MeHgCl (1 nM) on the mtDNA Copy Number in EBs

The relative mtDNA content was calculated as 2 × 2^ΔCT^, where ΔCT was based on the formula (ΔCT = nDNA(HBB)_CT_ − mtDNA(ND1)_CT_) [59]. In EBs cultured in 21% O_2_, the MeHgCl (1 nM µM) mtDNA content decreased significantly (*p* < 0.5) from 9227 ± 2638 to 5534 ± 2046. No significant changes in the mtDNA copy number were observed in EBs grown in 5% O_2_ in the presence of 1 µM MeHgCl. We observed a significant difference in this variable between EBs grown at 21% O_2_ with MeHgCl and EBs treated with MeHgCl at 5% O_2_ (*p* < 0.001). EBs cultured under control conditions (without MeHgCl) under atmospheric and physioxia conditions also differed significantly (*p* < 0.001). The mtDNA content was significantly higher in EBs cultured in 21% O_2_ than in EBs cultured in 5% O_2,_ regardless of the MeHgCl concentration (*p* < 0.001) (Figure 5B). The results are summarized in Appendix A.

#### 3.5.2. The Effect of Different Oxygen Concentrations and a Low Dose of MeHgCl (1 nM) on DNA Damage in EBs

Among numerous types of oxidative DNA damage, apurinic/apyrimidinic (AP or abasic) sites are one of the prevalent lesions of oxidative DNA damage [61]. EBs cultured under both tested oxygen conditions without MeHgCl were characterized by similar AP site levels. Exposure of EBs to MeHgCl (1 nM) resulted in an increase in the AP site levels in EBs grown in 21% O_2_ from 8.09 ± 1.04 to 9.38 ± 0.53 (*p* < 0.05). EBs were cultured with MeHgCl for 14 days before DNA damage at AP sites. The level of the AP site did not change significantly in EBs cultured in 5% O_2_ after exposure to a low dose of MeHgCl (Figure 5C).

### 3.6. The Effect of Different Oxygen Concentrations and a Low Dose of MeHgCl (1 nM) on the Differentiation of EBs into Three Germ Layers

The schedule of the experimental design to assess the effects of oxygen (21% O_2_, 5% O_2_) and a low concentration of MeHgCl (1 nM) on the differentiation potential of EBs into three germ layers is shown in Figure 6A. The results are summarized in Appendix A.

To test the effects of MeHgCl and oxygen conditions on the differentiation of EBs, the EBs were exposed to MeHgCl for 14 days in 21% and 5% oxygen. After this period, pluripotency (Oct4, Nanog) and three germ layer formation markers (ectoderm/neuroectoderm: Pax6, Nestin, NF-200, endoderm: FOXA2, mesoderm: αSMA) were detected. Furthermore, in EBs cultured in both oxygen conditions with and without a low dose of MeHgCl, the markers of proliferation (Ki67), DNA damage (H2AX), and apoptosis (casp3) were visualized. The protein expression of these markers was analyzed by immunocytochemistry and visualized with confocal microscopy (Figure 6B).

The embryoid bodies that developed in 21%O_2_ during a chronic (14-day) exposure to 1 nM MeHgCl revealed a smaller size (Figure 6B) and contained strongly fragmented nuclei compared to the control without treatment. A smaller number of Ki67-positive cells further confirmed the disturbance of EBs proliferation in EBs cultured under MeHgCl conditions (Figure 6B). In both groups, a lack of Oct4 was observed. In the control groups, endoderm (FOXA2), mesoderm (αSMA), and ectoderm markers (Nestin, Pax6, NF-200) were detected. However, ectoderm markers were expressed more strongly than endoderm and mesoderm markers. After MeHgCl treatment, we observed a decrease in the early neuroectodermal marker Pax66 and the neuronal marker NF200. Nestin, a neural stem cell marker, was still present in the “neural rosettes” parts of the EBs. MeHgCl disturbed the formation of all three germ layers in the EBs. After MeHgCl treatment, a NANOG pluripotency marker with cytoplasmic localization was detected. We did not observe a decisive difference in DNA damage in the EBs treated with MeHgCl compared to the control. The presence of H2AX foci indicated DNA damage in EBs. Qualitative immunocytochemical analysis showed differences between EBs developed under different oxygen conditions. It was easy to see that the development of EBs was slower (delayed) in EBs grown in 5% O_2_ than in EBs cultured in 21% O_2_. The expression of all three germ layers was lower in EBs developed in 5% O_2_. Furthermore, the cytotoxic effect was more robust in EBs grown in 21% O_2_. A low dose of MeHgCl (1 nM) more frequently resulted in cell nuclei, chromatin condensation, DNA fragmentation, and nuclear envelope collapse in EBs cultured in 21% O_2_ than in EBs cultured in 5% O_2_. We showed that the lethality of EBs strongly depended on the oxygen concentration in which EBs treated with MeHgCl were developed.

Function prediction of genes encoding the analyzed proteinsand network analysis (Appendix A) were performed to understand better the biological function of the analyzed ICC proteins. For this reason, the proteins analyzed in the ICC were divided into three groups take an account genes encoding the analyzed proteins. *POU5F1* and *NANOG* were included in the first group of genes. These proteins are transcription factors that play an essential role in maintaining pluripotency, embryonic morphogenesis, formation of the primary germ layer, the stem cell population maintenance, and cell number maintenance. Furthermore, *POU5F1* and *NANOG* regulate gene expression patterns by gene silencing and play an essential role in regulating transcription (DNA binding) and cell fate specification. Network in which *POU5F1* and *NANOG* participated based mainly on physical interactions. Other interactions identified in the network were related to gene co-expression, colocalization, genetic interactions, pathways, or shared protein domains. The second group of genes functions was associated with epithelial cell differentiation, including *ACTA2, PAX6*, and intermediate filament cytoskeleton (*NES, NEFH*). *FOXA2*, along with *POU5F1* and *PAX6*, plays an essential role in regulatory region DNA binding, and *POU5F1* participates in cell fate specification.

Furthermore, *FOXA2* and *PAX6* played an essential role in determining the fate of neurons. The larger group of interactions was associated with co-expression in the analyzed network. The second group of interactions was interactions related to pathways. The smallest group of interactions was based on sharing protein domains and genetic interactions.

The last of the analyzed networks (Appendix A) shows genes encoding the analyzed proteins related to the response to DNA damage (*MKi67, CASP3, H2AX*), which are involved in histone H2AX phosphorylation, proliferation (*MKi67*), and apoptosis (*CASP3*). The primary function of H2AX is related to the regulation of the response to the stimulus of DNA damage, regulation of DNA repair, and changes in DNA confirmation. Furthermore, the H2AX protein plays an important role is essential in DNA integrity and cell cycle checkpoints. At the same time, the *CASP3* function is associated mainly with apoptotic nuclear changes and the apoptosis execution phase. In the gene network, the primary type of interaction was physical interaction. Other interactions were associated with co-expression, colocalization, genetic interactions, and interactions based on shared pathways or protein domains.

## 4. Discussion

The research in vitro model used in this study—hiPSC-derived embryoid bodies (EBs)—is widely accepted as an ethical alternative to hESC-derived models [32,33]. EBs are three-dimensional aggregates formed by PSCs grown in suspension [15]. During EBs development, the formation of the three embryonic germ layers (ectoderm, mesoderm, and endoderm) occurs [19,20,21], which provides an opportunity to evaluate the effects of different compounds on cell fate decisions and behavior during human embryo development [62,63,64]. On day 13–14 of human embryo development, the primitive streak is formed posterior to the epiblast, creating three germ layers of the embryo proper [24,25,26]. For this reason, we evaluated the teratogenic and genotoxic effects of MeHgCl under different oxygen conditions in 14-day-old hiPSC-derived EBs. **To our knowledge, this is the first report presenting an analysis of the role of oxygen conditions in the teratogenic effect of MeHgCl exposure in the EB in vitro model.**

The number and activity of mitochondria are important factors that play a critical role in the cellular response to oxidative stress [65]. Mitochondria are the primary source of ROS, while ROS are one of the primary sources of DNA damage [66]. During differentiation, mitochondrial activity and ROS generation increase, which is accompanied by changes in mitochondria shape [67,68]. Our previous study showed that hiPSCs grown under 21% O_2_ conditions had more elongated mitochondria and were characterized by a higher level of ROS than hiPSCs grown under physioxia. Five-day hiPSC exposure to MeHgCl increased ROS accumulation only in hiPSCs cultured in 21% O_2_ at all tested doses, which was not the case in hiPSCs grown in 5% O_2_ [29]. Moreover, in 5% O_2_, hiPSCs showed elevated expression of HIF2α and HIF3α in the cytoplasm and started expressing this factor at low levels in the nucleus, which was not observed in hiPSCs grown in 21% O_2_. Surprisingly, hiPSCs under physioxia conditions were characterized by increased expression of genes involved in regulating mitochondrial biogenesis: *TFAM* and *POLG1* [29]. In the current study, short exposure (24 h) of hiPSCs cultured in 21% O_2_ to MeHgCl increased the number of apoptotic cells, increased ROS accumulation, and decreased the mitochondrial membrane potential only at the highest dose, which did not have an adverse impact on hiPSCs viability. Hence, 21% O_2_ was confirmed to increase ROS accumulation in hiPSCs treated with MeHgCl.

Immunocytochemical analysis of control EBs grown in 21% O_2_ confirmed a higher expression of Nestin (ectoderm), FOXA2 (endoderm), and αSMA (mesoderm) markers. Neuroectodermal markers (Pax6, Nestin, and NF-200) were more strongly expressed under both oxygen conditions than mesoderm (αSMA) and endoderm (FOXA2) markers. In addition, the *NES* and *SOX17* genes were more strongly expressed in EBs developed in 21% O_2_. In 21% O_2_, EBs were characterized by higher expression of Pax6, Nestin, and NF-200 than EBs developed in 5% O_2_. The expression of the pluripotency markers Oct4 and NANOG was negatively regulated by MeHgCl in all EBs regardless of the culture conditions, which can be explained by the fact that downregulation of Oct4 and NANOG is essential for the initiation of hiPSCs differentiation. MeHgCl decreased the relative expression of the endoderm marker (*SOX17*) and increased the mesoderm marker (*TBXT*) in 21% O_2_. The expression of *NES* did not change in MeHgCl-treated EBs under high oxygen conditions. However, we observed a more potent cytotoxic effect of MeHgCl at 21% O_2_. An increased differentiation rate was observed under atmospheric oxygen conditions, while the expression of three germ layer genes (*NES, SOX17, TBXT*) was not altered in EBs treated with MeHgCl at 5% O_2_. Earlier studies showed that 21% O_2_ could increase the activity of mitochondrial OxPhos, increase ROS production, and cause spontaneous differentiation in hiPSCs and hESCs [43]. These data suggest that 21% oxygen conditions promoted differentiation into the three germ layers and mainly supported neural differentiation of control EBs. On the other hand, MeHgCl reduced the ability of hiPSCs cultured in 21% O_2_ to form EBs and to enhance their development, while such effect was not observed in physioxia conditions (5% O_2_). Taking into consideration these results, we hypothesize that **DNA damage could cause premature differentiation of stem cells to protect their genome integrity.** This hypothesis is supported by the results of research conducted by other authors [51].

In our previous study, 5-day of exposure of hiPSCs to MeHgCl increased the number of mtDNA copies in both oxygen (21%, 5%) conditions, which was potentially related to the effect of compensation; however, upregulation of the mtDNA content in 5% O_2_ was lower than in hiPSCs grown in 21% O_2_ [29]. The opposite impact of MeHgCl on mtDNA content was observed in 3D culture of EBs in this report. Long-term EB exposure to a low dose of MeHgCl decreased the mtDNA content only in 21% O_2_ environment. This effect was not observed in EBs developed in 5% O_2_. Surprisingly, we recorded a massive difference in mtDNA content between EBs grown without MeHgCl in 21% and 5% oxygen. In EBs developed at 5% O_2_, mtDNA copy numbers were found to be similar to those observed in hiPSCs cultured in both oxygen conditions [29]. In addition, EBs cultured in 21% O_2_ were characterized by an increased quantity of three germ layers formation markers, which co-existed with a higher number of mtDNA copies. The increased number of mtDNAs observed in EBs in 21% O_2_ suggests that EBs use different ATP generation methods when cultured under different oxygen conditions and the metabolism in 21% O_2_ promotes the formation of oxygen-free radicals, sensitizing EBs to MeHgCl exposure.

The main part of this study was focused on the influence of MeHgCl on the regulation of the expression of the genes involved in DNA repair and how the consequences of these changes were involved in the teratogenicity of MeHgCl regarding the oxygen level. Studies of other groups have shown that low-oxygen conditions (hypoxia) reduced the expression of DNA repair genes in cancer cells [46,47,48,49]. Our results indicated that the expression of DNA repair genes in EBs depended on the oxygen level; however, we observed the opposite effect of low oxygen as compared to the results obtained in cancer cells. **EBs grown in low-oxygen conditions were characterized by increased expression of DNA repair genes such as *OGG1, POLG1, PARP1,* and *ATM* compared to EBs grown in 21% O_2_. High expression of DNA repair genes, which are responsible for the integrity of the embryo genome and protects it from environmental compounds’ genotoxic and teratogenic effects, was recorded only in physioxia.**

Oxidative DNA damage is repaired primarily by the base excision repair pathway, which is initiated by oxoguanine glycosylase 1 (*OGG1*) in mammals. *OGG1* removes the damaged base and generates an apurinic/apyrimidinic (AP) site [69]. In our study, we demonstrated a decreased expression of the *OGG1* gene in MeHgCl-treated EBs, which is the evidence for the ability of MeHgCl to inhibit DNA repair in the early stages of the three germ layers formation. The OGG1 glycosylase, for proper and efficient functioning, requires PARP1 [70,71]. In our study, a downregulation of *OGG1* and *PARP1* expression was observed in EBs treated with a low dose of MeHgCl, but only in physioxia (5% O_2_) conditions. **In that case, the reduction in *OGG1* and *PARP1* expression was not associated with increased DNA damage (AP sites) but might have been the reason for the increased sensitivity of EBs to other oxidative DNA damage agents.**

The results of our research support the observations of other authors who showed a reduced expression of BER-associated genes, including glycosylase, an apurinic/apyrimidinic endonuclease, and DNA ligase after exposure to MeHg [72]. Other researchers also showed that at sub-cytotoxic concentrations all three mercury species strongly disturbed poly(ADP-ribosyl)ation, a signaling reaction induced by DNA strand breaks [73].

DNA polymerase gamma (Polγ, POLG1) is a nuclear-encoded mitochondrially active DNA replication and repair enzyme [74,75], which is essential for mammalian embryogenesis [76]. Homozygous knockout of Polγ in mice caused early developmental defects leading to embryonic lethality [76]. Our previous studies [29] showed that the treatment of hiPSCs with MeHgCl was associated with a strong downregulation of *POLG1* expression under both oxygen conditions. A reduction in *POLG1* expression was shown in EBs developed in both 21% and 5% oxygen conditions. **Considering the obtained results, we postulate that the downregulation of *POLG1* gene expression is one of the most critical mechanisms of MeHgCl toxicity, regardless of the oxygen conditions.**

Another master player of the DNA damage response (DDR) processes is Ataxia-telangiectasia mutated (ATM), which coordinates a complex network of signaling cascades, including cell cycle checkpoints and DNA double-strand breaks, to maintain genomic integrity [77,78,79]. ATM is necessary to initiate double-strand break (DSB) repair by homologous recombination (HR) [77,78], which acts as a genotoxic stress transducer and teratological suppressor tole to protect the embryo from pathological cell death and teratogenesis initiated by DNA damage. Consequently, ATM may have a significant developmental role in protecting the embryo from more subtle oxidative DNA damage caused by endogenous and xenobiotic-enhanced ROS [80,81,82]. Our study showed that the *ATM* gene was more highly expressed in EBs cultured in low-oxygen conditions. In EBs developed in atmospheric (21% O_2_) and physioxia (5% O_2_) conditions, MeHgCl did not adversely affect the expression of the *ATM* gene. Due to the higher level of *ATM* expression in EBs grown in physioxia, better protection against DNA damage and better DNA damage repair involving *ATM* gene pathway is potentially characteristic of EBs cultured in 5% O_2_ rather than in 21% O_2_. PSCs express higher levels of HR factors than somatic cells, which supports maintaining their genome integrity despite fast cell cycle progression. High expression of HR factors could accelerate differentiation, and they can decrease the elevated rate of DNA breaks by repairing the breaks efficiently and rapidly [11,12,13]. Our results have shown that EBs grown in 21% O_2_ revealed more advanced differentiation than in 5% O_2_, and their expression level of DNA repair genes was lower. Such lower expression of DNA repair genes was also the case for more differentiated cells as compared to PSCs [51].

Mitochondrial transcription factor A (TFAM) is a core mitochondrial transcription factor [83,84]. *TFAM* knockout mice showed embryonic lethality, thereby indicating that TFAM plays a critical role in embryonic development [85]. Furthermore, some evidence indicates that *TFAM* may play a role in mtDNA repair regulation [86,87] by degrading damaged mtDNA that contains AP sites [88], which are ubiquitous DNA lesions and important intermediates during BER. In our study, the upregulation of *TFAM* expression was observed in EBs cultured under physioxia conditions and treated with MeHgCl. However, in 5% O_2_, the changes in *TFAM* expression did not significantly affect mtDNA copy number in MeHgCl-treated EBs. EBs grown in 21% O_2_ without MeHgCl showed an increased expression of *TFAM* when compared to the EBs cultured in 5% O_2_. This suggests important role of *TFAM* in protection of mitochondrial genome integrity in physioxia conditions; however, it needs more investigations.

Mitophagy is the selective degradation of mitochondria by autophagy, which prevents the accumulation of dysfunctional mitochondria. This process often occurs in post-damage or post-stress defective mitochondria [55,89]. MeHg can alter mitochondrial function and viability, decrease mitophagy and autophagy, and increase oxidative stress [54]. PARK2 is one of the primary regulators of mitophagy [89]. In hiPSCs, MeHgCl caused the downregulation of *PARK2* gene expression in an oxygen-dependent manner. *PARK2* expression was more extensively downregulated in hiPSCs cultured in 21% O_2_ than in those cultured in 5% O_2_ [29]. We also observed downregulation of the expression of the *PARK2* gene in EBs grown with MeHgCl at 21% O_2_. This effect was not detected in EBs cultured in physioxia conditions. We showed that atmospheric oxygen enhanced the effect of MeHgCl on EBs by increasing the downregulation of *PARK2* expression. Higher expression of the *PARK2* gene was observed in EBs developed in 5% O_2_ than in 21% O_2_. Therefore, we suggest that EBs grown in physioxia conditions present a more effective system to **prevent cells from the accumulation of dysfunctional mitochondria.**

One of the postulated mechanisms of MeHgCl genotoxicity involving microtubules is the inhibition of mitotic spindle formation and chromosome segregation [90,91] by influencing microtubules dynamics. Our research results support this hypothesis showing that expression of the *TUBB3* gene (encoding the mitotic spindle-building protein Tubulin Beta 3 Class III) was upregulated in EBs treated with MeHgCl in 21% O_2_.

EBs grown at physioxia concentrations revealed increased expression of DNA repair genes, thus the mechanism of DNA damage repair was more efficient. The consequence of this fact was a lower level of AP sites in EBs growing in 5% O_2_. Our data revealed a greater sensitivity of EBs to DNA oxidative damage caused by long-term exposure to a low dose of MeHgCl at 21% O_2_. Single cells with H2AX foci were detected in EBs in all tested conditions, but significant upregulation of the number of AP sites was observed only at 21% O_2_ after MeHgCl treatment. This phenomenon coexisted with the higher expression of the apoptosis marker casp3. Moreover, high oxygen EBs treated with MeHgCl contained cells that revealed nuclear shrinkage with chromatin condensation and blabbing of the plasma membrane, with DNA and nuclear fragmentation, which were observed more often than in 5% O_2_ [92].

In conclusion, our results indicate that atmospheric oxygen (21%) conditions during EBs culture increased the sensitivity of EBs to MeHgCl. The consequences of the development of EBs in 21% O_2_ in the presence of MeHgCl were related to increased oxidative damage of DNA, DNA fragmentation, and altered expression of genes and proteins involved in forming the three germ layers. We also observed differences in mtDNA copy number between EBs cultured in different oxygen conditions. **In this study, we confirmed that physioxia (5% O_2_) better mimics the environment of the human embryo development and protects EBs from MeHgCl toxicity by increasing the expression of genes involved in DNA repair and mitophagy compared to the atmospheric oxygen condition.**

## Figures and Tables

**Figure 1 cells-12-00390-f001:**
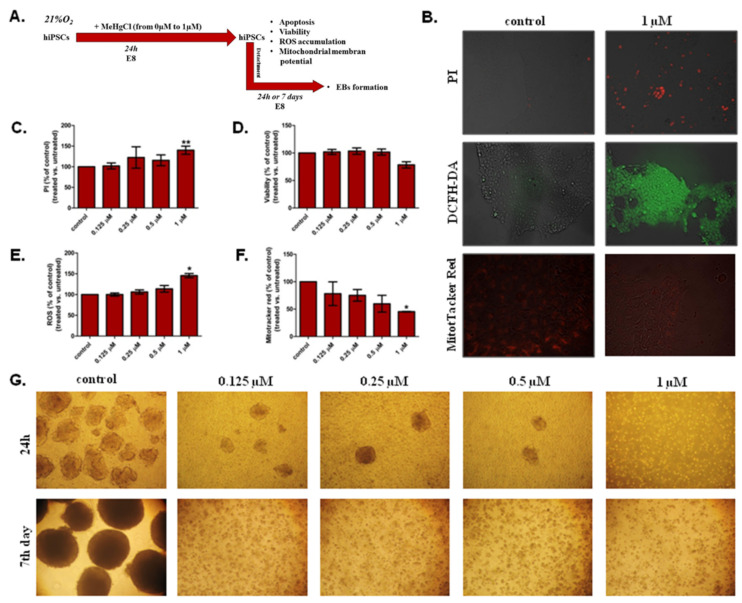
The effect of methylmercury chloride (MeHgCl) on human induced pluripotent stem cells (hiPSCs) culture and embryoid bodies (EBs) growth at 21% O_2_. (**A**). The schedule of the experimental design to assess the toxicity of different doses of MeHgCl for hiPSCs growing in a monolayer. (**B**). Fluorescence microscopy qualitative visualization of apoptotic cells (propidium iodide (PI), ROS accumulation (DCFH-DA), and mitochondrial membrane potential (Δψ m) (MitoTracker Red)) in hiPSCs untreated and treated with 1 µM MeHgCl for 24 h. Estimation of: (**C**). Cell death; (**D**). Cell viability; (**E**). Reactive oxygen species (ROS) accumulation and (**F**). Mitochondrial membrane potential (Δψ m) in iPSC grown in monolayer. The results are presented as the mean (± SD). The bars on the chart show the comparison between hiPSCs and hiPSCs treated with MeHgCl (Kruskal-Wallis test with Dunn’s multiple comparison test, * *p* < 0.05; ** *p* < 0.01). hiPSCs were exposed to MeHgCl for 24 h (0–1 µM). The results are a percentage (%) of the untreated control. (**G**). The ability to form EBs from hiPSCs after MeHgCl treatment. Before EBs generation from hiPSCs, hiPSCs were treated with MeHgCl (21% O_2_) at doses from 0 µM to1 µM for 24 h. After this time, an EBs formation assay was performed. The formation and development of EBs from hiPSCs were performed in MeHgCl-free medium. The hiPSCs ability to EBs formation after MeHgCl exposure was analyzed by light microscopy 24 h and 7 days after EBs generation.

**Figure 2 cells-12-00390-f002:**
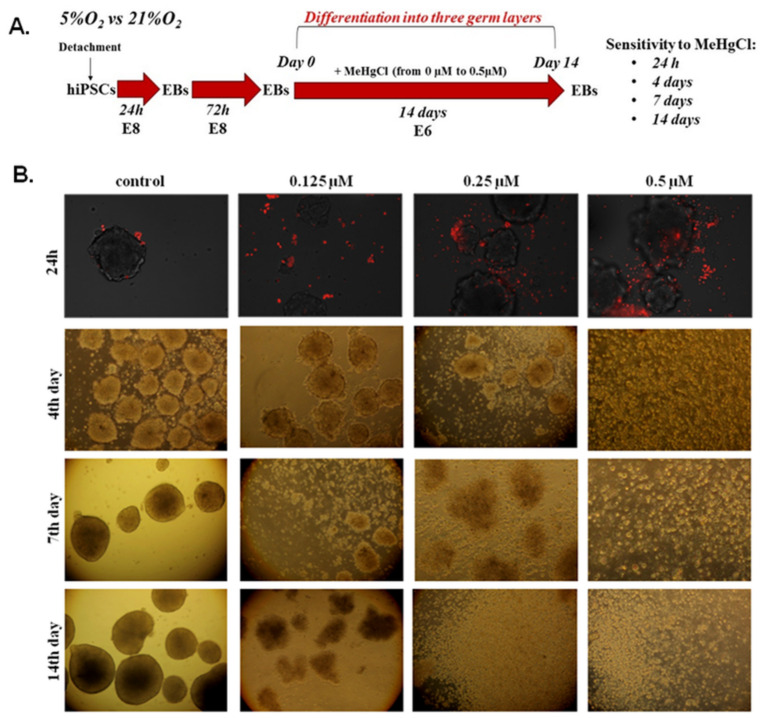
Dose- and time-dependent sensitivity of EBs to MeHgCl. (**A**). The schedule of the experimental design. (**B**). The effect of exposure of EBs to different doses of MeHgCl in atmospheric oxygen conditions (21% O_2_). The sensitivity of EBs to MeHgCl was analyzed after different exposure time points: 24 h (fluorescence microscopy) and 4, 7, and 14 days (light microscopy). EBs grew in 21% O_2_ oxygen with MeHgCl (in noncytotoxic for hiPSC doses ranging from 0 µM to 0.5 µM). Apoptotic cells (PI) were visualized with fluorescence microscopy after 24 h of EBs exposure to MeHgCl.

**Figure 3 cells-12-00390-f003:**
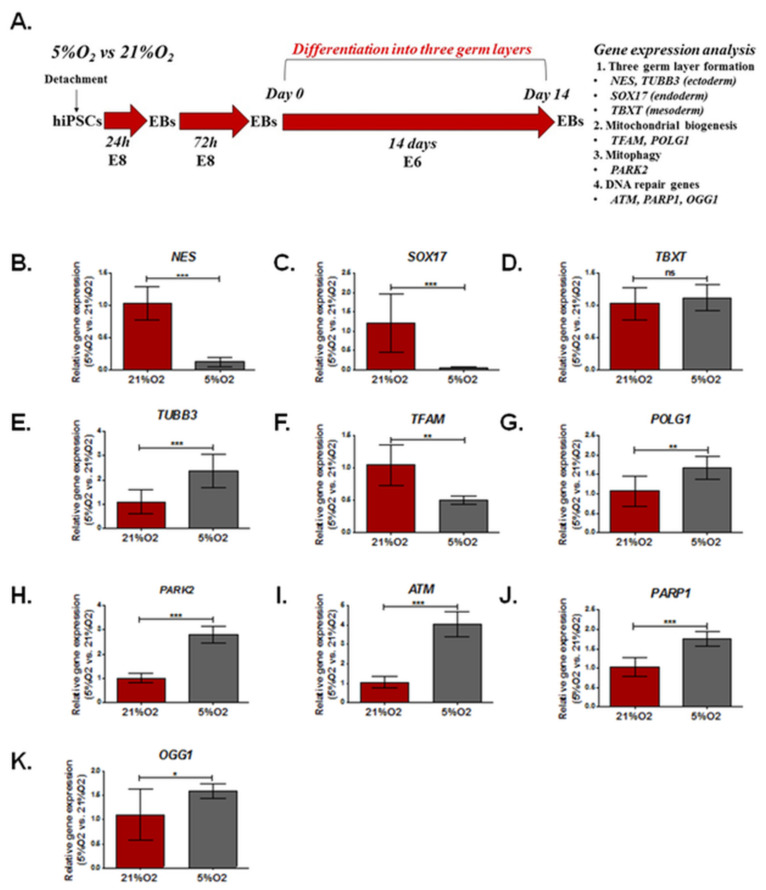
Relative gene expression in EBs cultured at different oxygen concentrations (21% O_2_, 5% O_2_). (**A**). The schedule of the experimental design. Relative gene expression was compared between EBs grown in 5% O_2_ and 21%O_2_. The impact of oxygen concentration (21% O_2_, 5% O_2_) on the relative expression of genes associated with the formation of three germ layers. (**B**). *NES* (ectoderm, neuroectoderm). (**C**). *SOX17* (endoderm). (**D**). *TBXT* (mesoderm). (**E**). *TUBB3* (neuroectoderm). Mitochondrial biogenesis and mtDNA replication (**F**). *TFAM* and (**G**). *POLG1*. *POLG1* is also involved in mtDNA damage repair). (**H**). *PARK2* (mitophagy). (**I**). *ATM*, (**J**), *PARP1,* and (**K**) *OGG1* (DNA repair) in EBs. The results are presented as the mean (± SD). The brackets show statistically significant differences between EBs cultured in 21% and 5% oxygen concentrations for 14 days in Essential E6 medium that induce the spontaneous formation of three germ layers, as determined by the Mann-Whitney test (* *p* < 0.05, ** *p* < 0.01, *** *p* < 0.001).

**Figure 4 cells-12-00390-f004:**
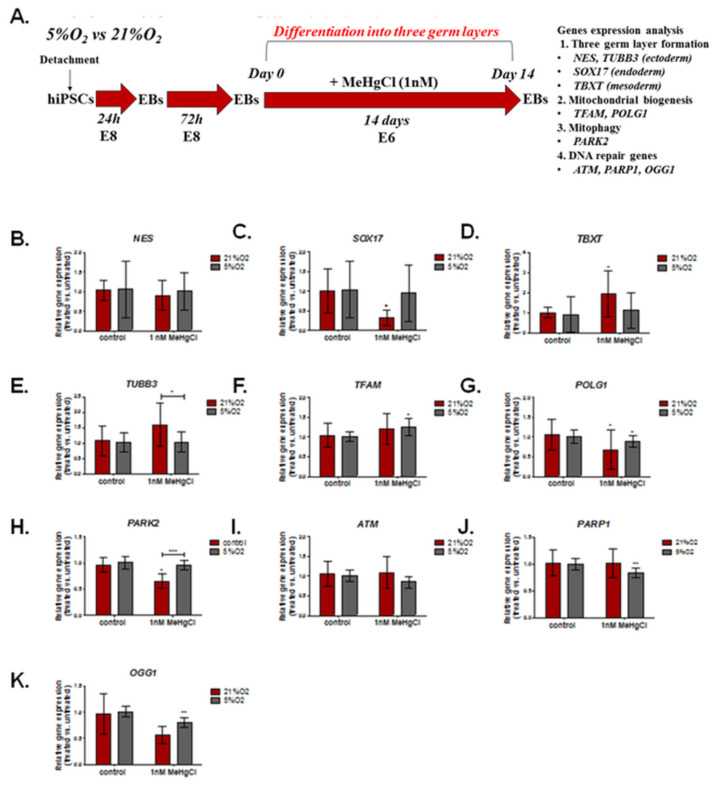
Relative expression of selected genes in EBs treated with MeHgCl vs. EBs untreated with MeHgCl in 21% O_2_ and 5% O_2_. EBs were exposed to MeHgCl (1 nM) for 14 days. The results in brackets represent the mean (± SD). (**A**). The schedule of the experimental design. The impact of oxygen concentration (21% O_2_, 5% O_2_) on the relative expression of genes associated with the formation of three germ layers: (**B**). *NES* (ectoderm, neuroectoderm). (**C**). *SOX17* (endoderm). (**D**). *TBXT* (mesoderm). (**E**). *TUBB3* (neuroectoderm). Mitochondrial biogenesis and mtDNA replication (**F**). *TFAM* and (**G**). *POLG1*. *POLG1* is also involved in mtDNA damage repair). (**H**). *PARK2* (mitophagy). (**I**). *ATM*, (**J**), *PARP1*, and (**K**) *OGG1* (DNA repair) in EBs. The figure presents significant differences between EBs exposed to MeHgCl (1 nM) in 21% and 5% oxygen (* *p* < 0.05, ** *p* < 0.01, *** *p* < 0.001), as determined by two-way ANOVA with Bonferroni’s test. Brackets show statistically significant differences between EBs treated with MeHgCl and untreated EBs cultured under 21% O_2_ and 5% O_2_ conditions determined by the Mann-Whitney test (* *p* < 0.05, ** *p* < 0.01, *** *p* < 0.001).

**Figure 5 cells-12-00390-f005:**
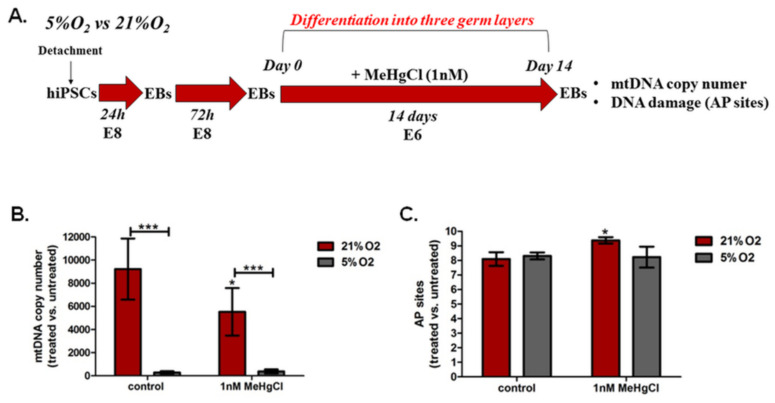
mtDNA copy number and DNA damage in EBs treated with MeHgCl vs. EBs untreated with MeHgCl in 21% O_2_ and 5% O_2_. (**A**). The schedule of the experimental design. Effects of oxygen (21% O_2_, 5% O_2_) and a low MeHgCl concentration (1 µM) on: (**B**). mtDNA copy number and (**C**). DNA damage (the level of apurinic/apyrimidinic) sites). The analysis was carried out in EBs after 14 days of exposure to 1 nM MeHgCl under 21% O_2_ and 5% O_2_. The figure presents significant differences between EBs exposed to MeHgCl (1 nM) in 21% and 5% oxygen (* *p* < 0.05, *** *p* < 0.001), as determined by two-way ANOVA with Bonferroni’s test. The brackets are presented as the mean (± SD). Significant differences between MeHgCl-treated EBs and untreated EBs cultured under 21% or 5% oxygen conditions were determined using the Mann-Whitney U test (* *p* < 0.05, *** *p* < 0.001).

**Figure 6 cells-12-00390-f006:**
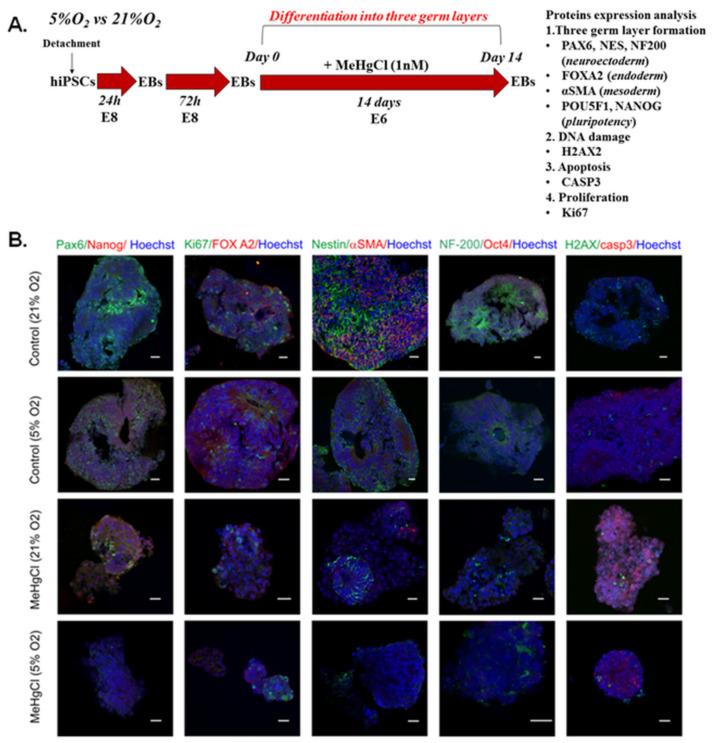
The effects of oxygen (21% O_2_, 5% O_2_) and a low concentration of MeHgCl (1 nM) on the differentiation potential of EBs into three germ layers. (**A**). The schedule of the experimental design. (**B**). Fluorescent images of the expression of proteins associated with pluripotency (Oct4, Nanog); neuroectoderm formation (Pax6, Nestin, NF-200); mesoderm (αSMA); endoderm (FOXA2); proliferation (Ki67); DNA damage (H2AX); and apoptosis (casp3) in EBs grown in different oxygen conditions. Scale bar: 20 µM.

**Table 1 cells-12-00390-t001:** Primers used for qPCR *.

Gene Symbol	GenBank Number	Primer Sequence	Amplicon Length (bp)
*HBB F*	NG_059281.1	GCTCGGTGCCTTTAGTGATG	136
*HBB R*		ACATCAAGCGTCCCATAGAC	
*ND1 F*	DQ092356.1	TACGGGCTACTACAACCCTTC	77
*ND1 R*		ATGGTAGATGTGGCGGGTTT	

* F—Forward, R—Reverse.

**Table 2 cells-12-00390-t002:** Primers sequences used for RT-qPCR *.

Primers	GenBank Number	Primer Sequence	Amplicon Length
*ATMF*	NM_001351834.2	AGGAATCACTGGATCGCTGTC	100
*AMTR*		CGTGAACACCGGACAAGAGT	
*OGG1F*	NM_002542.6	AGACCAACAAGGAACTGGGAAAC	81
*OGG1R*		CACTGAACAGCACCGCTTGG	
*PARP1F*	NM_001618.4	CAGCTTCTGGAGGACGACAA	108
*PARP1R*		CTCCTTGGACGGCATCTGTT	
*TFAM F*	NM_005011.4	TGAAAGATTCCAAGAAGCTAAGGGT	132
*TFAM R*		TAACGAGTTTCGTCCTCTTTAGCAT	
*POLG1 F*	NM_002693.3	GGCATTGTTGCTTGTTGGGT	144
*POLG1 R*		TTTCCCCTTCTAGGGCACTG	
*PARK2 F*	NM_004562.3	TCCCAGTGGAGGTCGATTCT	105
*PARK2 R*		CCTGCGAAAATCACACGCAA	
*SOX17F*	NM_022454.4	TGGACCGCACGGAATTTGAA	101
*SOX17R*		GCTGTCGGGGAGATTCACAC	
*NES F*	NM_006617.2	CCCCGTCGGTCTCTTTTCTC	96
*NES R*		TCGTCTGACCCACTGAGGAT	
*TBXT F*	NM_003181.4	CGATCCTGGGTGTGCGTAA	100
*TBXT R*		CCGATGCCTCAACTCTCCAG	

* F—Forward, R—Reverse.

**Table 3 cells-12-00390-t003:** Primers sequences used for RT-qPCR reference gene validation.

Primers	GenBank Number	Primer Sequence	Amplicon Length
*ACTB F*	NM_001101.3	GCTCACCATGGATGATGATATCGC	169
*ACTB R*		CACATAGGAATCCTTCTGACCCAT	
*EEF1A1 F*	NM_001402.5	TGTTCCTTTGGTCAACACCGA	122
*EEF1A1 R*		ACAACCCTATTCTCCACCCA	
*EID2 F*	NM_153232.3	GGCATCGCTCTGTCCAGTTA	74
*EID2 R*		GCTTGGACATCTCAGACCGT	
*GAPDH F*	NM_002046.5	GTTCGACAGTCAGCCGCATC	90
*GAPDH R*		TCCGTTGACTCCGACCTTCA	
*RPLP0 F*	NM_001002.3	CCTCGTGGAAGTGACATCGT	76
*RPLP0 R*		CTGTCTTCCCTGGGCATCAC	
*TBP F*	NM_003194.4	GCAAGGGTTTCTGGTTTGCC	80
*TBP R*		CAAGCCCTGAGCGTAAGGTG	

**Table 4 cells-12-00390-t004:** Primary antibodies used for EBs immunofluorescence staining.

Antibodies	Catalog Number	Company	Dilution
anti-OCT4, (rabbit/IgG)	PA5-27438	Invitrogen, Thermo Fisher Scientific, Waltham, MA, USA	1:1000
anti-NANOG, (rabbit/IgG)	PA1-097	Invitrogen, Thermo Fisher Scientific, Waltham, MA, USA	1:200
anti-PAX6, (mouse/IgG1)	Ma1-109	Invitrogen, Thermo Fisher Scientific, Waltham, MA, USA	1:200
anti-NESTIN, (mouse/IgG1)	Mab5326	Sigma-Aldrich, Merck KGaA, Darmstadt, Germany	1:500
anti-NF200, (mouse/IgG1)	N0142	Sigma-Aldrich, Merck KGaA, Darmstadt, Germany	1:200
anti-αSMA, (mouse/IgG2a)	A2547	Sigma-Aldrich, Merck KGaA, Darmstadt, Germany	1:600
anti-FOXA2, (mouse/IgG2aκ)	WH0003170M1	Sigma-Aldrich, Merck KGaA, Darmstadt, Germany	1:300
anti-Histone H2A.X phosphorylated at Ser139, (mouse/IgG1)	05-636	Sigma-Aldrich, Merck KGaA, Darmstadt, Germany	1:100
anti-CASP3, (rabbit/IgG)	9664	Cell Signaling Technologies, Danvers, MA, USA	1:1000
anti-Ki67, (rabbit/IgG)	AB9260	Sigma-Aldrich, Merck KGaA, Darmstadt, Germany	1:1000

**Table 5 cells-12-00390-t005:** Secondary antibodies used for EBs immunofluorescence staining.

Antibodies	Catalog Number	Company	Dilution
Alexa Fluor 488-conjugated goat anti-rabbit IgG (H+L) cross-adsorbed secondary antibody	A11008	Invitrogen, Thermo Fisher Scientific, Waltham, MA, USA	1:1000
Alexa Fluor 546-conjugated goat anti-rabbit IgG (H+L) highly cross-adsorbed secondary antibody	A11035	Invitrogen, Thermo Fisher Scientific, Waltham, MA, USA	1:1000
Alexa Fluor 488-conjugated goat anti-mouse IgG1 cross-adsorbed secondary antibody	A-21121	Invitrogen, Thermo Fisher Scientific, Waltham, MA, USA	1:1000
Alexa Fluor 546-conjugated goat anti-mouse IgG2a cross-adsorbed secondary antibody	A21133	Invitrogen, Thermo Fisher Scientific, Waltham, MA, USA	1:1000

## Data Availability

Not applicable.

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
