# Peer review of "Genes Involved in DNA Repair and Mitophagy Protect Embryoid Bodies from the Toxic Effect of Methylmercury Chloride under Physioxia Conditions"

_cells, 2023, doi:10.3390/cells12030390_

Round 1
Reviewer 1 Report
Here is my comments on the 4 subjects below about the case report entitled: "Genes involved in DNA repair and mitophagy protect embryoid bodies from the toxic effect of methylmercury chloride under physioxia conditions".
Literary quality:
The style and organization of writing, including paragraphing and flow, are clear and readable enough. The language of this study can be examined in terms of compliance with journal writing rules. It could be more fluid.
In this study, the author reported that "To our knowledge, this is the first report presenting an analysis of the role of oxygen conditions in the teratogenic effect of MeHgCl exposure in the EB in vitro model".
In this article, I did not find any over/under-reference, adequacy of sources, or excessive self-citation.
References have been duly used for this paper, and readers can access all kinds of information from these references if they wish. But, references should be carefully re-examined under journal guidelines.
The effects of Physioxia on stem cells have not been investigated for the first time.
However, the authors did not adequately cite these studies.
3) Quality of scholarship:
Topic: This study is important because it is the first report presenting an analysis of the role of oxygen conditions in the teratogenic effect of MeHgCl exposure in the EB in vitro model.
Title: The article title is compatible with the article content and provides information about it. But it has become a title as if it is talking about a new topic.
Abstract: The summary section is adequately summarized. It is consistent and sufficient within the scope of the research topic. It is at a level to present the necessary information to the reader.
Keywords: Keywords are sufficient and informative.
Introduction section: It is consistent and sufficient within the scope of the research topic. The authors have appropriately explained his suggestions and reservations. This section is adequate but too long and must summarize.
Material-methods section: The authors wrote the methods too long. but they didn't add a reference. The methods described are already known.
Results: It would be better if the results could be summarized for easier reading. The figures used are sufficient.
Discussion section: The authors compared their results with those of other authors and presented their recommendations.
But it's been too long. In this case, it may be less interesting. It is unnecessary to repeat the information already described in the findings.
4) Readership interest and decision:
It is a remarkable work in terms of the claims of the authors. In particular, the striking points should be highlighted to make the article more interesting and easy to read.
Author Response
"Please see the attachment."

Reviewer 2 Report
dear authors, the article is nice formulated and presented. I found only minor errors which have to be corrected.
Please, correct the citation in the Introduction “(Choi et al. 2020)” in number sa well sa add citation in the Dissussion page 21 “(cytacja)”. The references itself are quite old, čítača 2/3 are older than 5 yaers (1/2 is older than 10 years).
I would like to know why not all experiments were provided in 5% oxygen conditions sa well (I mean precultivation, the effect of MeHgCl on hiPSCs grown - subchapter 3.1, the effect of MeHgCl on the development of EBs - subchapter 3.2).
I would recommend to change the “low oxygen” to “physioxia” in the title of subchapter 2.3.
Author Response
"Please see the attachment."
